# Minimally Invasive Sampling Methods for Molecular Sexing of Wild and Companion Birds

**DOI:** 10.3390/ani13213417

**Published:** 2023-11-03

**Authors:** Maria-Carmen Turcu, Anamaria Ioana Paștiu, Lucia-Victoria Bel, Dana Liana Pusta

**Affiliations:** 1Department of Genetics and Hereditary Diseases, Faculty of Veterinary Medicine, University of Agricultural Sciences and Veterinary Medicine Cluj-Napoca, 400372 Cluj-Napoca, Romania; maria-carmen.turcu@usamvcluj.ro (M.-C.T.); dana.pusta@usamvcluj.ro (D.L.P.); 2New Companion Animals Veterinary Clinic, Faculty of Veterinary Medicine, University of Agricultural Sciences and Veterinary Medicine Cluj-Napoca, 400372 Cluj-Napoca, Romania; lucia.bel@usamvcluj.ro

**Keywords:** birds, molecular sexing, minimally invasive methods

## Abstract

**Simple Summary:**

Over 50% of birds are monomorphic, showing no sexually dimorphic traits, and in nestlings, the percentage is even higher. Early sex determination can be of great value in the management of wild birds, breeding of different bird species, improvement of breeding programs of captive birds, and evolutionary studies fields, and also for bird owners. In this study, we performed molecular sexing of wild and companion birds from various orders, such as *Falconiformes*, *Accipitriformes*, *Galliformes*, *Anseriformes*, *Passeriformes*, and *Psittaciformes*. Samples of oral swabs, feathers, and blood were collected from 43 birds. Conventional PCR was used to amplify the *CHD1-Z* and *CHD1-W* genes. The obtained results show that all types of samples can be used for molecular sexing of the studied species of birds. In conclusion, instead of using blood samples, our recommendation is to use minimally invasive samples (oral swabs and feathers) and test both types of samples on each bird.

**Abstract:**

Birds are highly social and must be paired in order to increase their welfare. Most bird species are monomorphic; therefore, molecular sexing helps provide appropriate welfare for birds. Moreover, early sex determination can be of great value for bird owners. The aim of this study was to demonstrate that sex identification in birds achieved using molecular methods and samples collected via minimally invasive methods is fast, efficient, and accurate. A total of 100 samples (29 paired samples of feathers and oral swabs and 14 tripled samples of feathers, oral swabs, and blood) from 43 birds were included in this study, as follows: wild birds (*Falconiformes*, *Accipitriformes*, landfowl—*Galliformes*, waterfowl—*Anseriformes*) and companion birds (*Passeriformes*, *Psittaciformes*—large-, medium-, and small-sized parrots). Amplification of *CHD1-Z* and *CHD1-W* genes was performed via conventional PCR. The results obtained from feathers were compared to those obtained from oral swabs and to those obtained from blood samples, where applicable. The obtained results show that all types of samples can be used for molecular sexing of all studied bird species. To the best of our knowledge, the present study reports, for the first time, molecular sex identification in Red Siskin (*Carduelis cucullata*) and Goldfinch (*Carduelis carduelis major*). For higher accuracy, our recommendation is to use minimally invasive samples (oral swabs and feathers) and to test both types of samples for each bird instead of blood samples.

## 1. Introduction

The origin of the present birds dates back to approximately 66 million years ago, when their ancestors survived a mass extinction event. They evolved and developed into a very large number of highly diverse species (more than 10,000), which are now spread all over the world. Despite their great phenotypic diversity, a large number of birds are monomorphic, with sexual dimorphism being absent (especially in chicks and juveniles) or hardly observable, even in some adults [1].

Sexual determinism in birds is chromosomal. Female birds are heterogametic with two distinct sexual chromosomes Z and W (ZW), while male birds are homogametic, presenting only the Z chromosome (ZZ). The *chromo-helicase DNA-binding protein gene* (*CHD1*), which is well-conserved and present in both sex chromosomes of all birds, allowed sex identification in the majority of avian species [2]. More specifically, sex identification is determined via PCR reaction markers used to amplify the homologous regions of the two genes *CHD1-Z* and *CHD1-W*. The differences between these two genes are demonstrated by the length polymorphism of introns. The amplification products are presented as a single copy of a gene for males (usually two Z chromosomes; the gene copy is usually of identical length), and in females, two copies are present due to the different length polymorphism of genes located in Z and W chromosomes [3]. Molecular sexing in birds aims to identify distinctive characteristics of the sex chromosomes Z and W. Genetic sexing is considered a noninvasive method compared to the classical sexing methods, like, for example, celioscopy. Moreover, the molecular method is considered safe, as collecting samples does not endanger the birds’ lives or expose them to risk of infection. Being able to collect feather samples, even molted feathers from the nest area, is an advantage for wild birds, allowing them to avoid the stress of handling. Plucking is generally considered to be a minimally invasive feather sampling method, whereas collecting feathers from birds that are molting is regarded as noninvasive [4,5,6]. Samples extracted from plucked feathers typically contain a higher quantity of DNA than those from molted feathers. Optimal DNA quality is obtained from freshly plucked contour or flight feathers with substantial rachis. In contrast, molted feathers yield lower-quality DNA, resulting in PCR success rates for sex determination of less than 50% [5,7]. Additionally, it has been observed that larger feathers harbor a higher DNA content than smaller feathers [5]. Therefore, it is advisable to collect multiple feathers, preferably a minimum of two, to ensure the reliability of the genetic analysis [7]. Birds of all ages can be safely sexed, especially juveniles or newly hatched birds, using oral swab samples [8,9,10,11,12] with the exception of birds that have oral lesions [13].

After fish, cats, and dogs, birds are the fourth most common pet in the US [14,15], while in the EU, birds rank as the third most popular pet [16,17]. In birds bred for more generations in captivity, such as Atlantic Canary (*Serinus canaria*), Budgerigars (*Melopsittacus undulatus*), Zebra Finches (*Taeniopygia guttata*), Lovebirds (*Agapornis* sp.), Cockatiels (*Nymphicus hollandicus*), etc., their behavior and physiology vary little from those of wild individuals [18]. Given their intricate social dynamics, birds thrive when paired, underscoring the significance of early molecular sexing in fostering their overall well-being [19].

Rarely, several avian species, including *Galliformes*, and specific landfowl, such as Common Pheasant (*Phasianus colchicus*), exhibit pronounced sexual dimorphism. Conversely, certain waterfowl species, like Whooper Swan (*Cygnus cygnus*), do not display visible sexual dimorphism, yet their sex can be determined by measuring the dimensions of their long bones [20].

*Psittaciformes*, commonly known as parrots, encompass approximately 400 species and represent some of the most popular pet birds [21,22]. While most parrot species lack visual sexual dimorphism, certain species may develop distinct sexual characteristics only after attaining sexual maturity. Scarlet Macaw (*Ara macao*), for instance, is classified into two subspecies, *Ara macao macao* and *Ara macao cyanopterus*, with juveniles exhibiting a plumage similar to that of adults yet lacking any apparent sexual dimorphism [22]. Similarly, both subspecies of African Grey Parrot (*Psittacus erithacus*), *Psittacus erithacus erithacus* and *Psittacus erithacus timneh*, exhibit no discernible sexual dimorphism [22]. African Gray Parrots, which fall under this category, attain sexual maturity between 3 and 5 years of age [22,23]. Sexual dimorphism manifests in different forms across various parrot species. For instance, White Cockatoo (*Cadatua alba*) exhibits sexual dimorphism in eye coloration. Both sexes possess a pale blue eye ring, while males showcase a dark brown iris, and females have a reddish iris. Additionally, females tend to have comparatively smaller heads and beaks than males [22]. In the case of the Rose-Ringed Parakeet (*Psittacula krameri*), only adult males possess a distinct black neck ring, while it is absent in females and in sexually immature birds of both sexes. This characteristic becomes apparent after the parakeets reach 3 years of age [22,23]. Similarly, Red-Rumped Parrot (*Psephotus haematonotus*) parrots exhibit sexual dimorphism, with males displaying bright green plumage and females exhibiting a dull olive coloration. The two subspecies, *P. h. haematonotus* and *P. h. caeruleus*, showcase further differences, including yellow abdomens and red rumps in males, while females maintain a uniform dull olive appearance [22]. Cockatiels demonstrate sexual dimorphism through their tail features, with males showcasing a long, uniformly gray, strongly graduated tail, while females and juveniles exhibit finely barred white tails with yellow barred gray outermost feathers [22]. The genus *Agapornis* encompasses nine species, with only three displaying sexual dimorphism. Breeding between these species adds complexity to sex determination due to their individual monomorphic phenotypes. Among these, Red-Headed Lovebird (*Agapornis pullarius*) is characterized by a small size, black inner wing face, and bright red head in males, contrasting with the completely green appearance of females. The Black-Winged Lovebird (*Agapornis taranta*), a larger species found at higher altitudes, features males with a black inner wing and red forehead and completely green females. The Grey-Headed Lovebird or Madagascar Lovebird (*Agapornis canus*) is considered the most primitive of the *Agapornis* genus, with males displaying gray head feathers and females appearing entirely green. In contrast, species such as *Agapornis roseicollis*, *Agapornis fischeri*, *Agapornis personatus*, *Agapornis swindernianus*, *Agapornis lilianae*, and *Agapornis nigrigenis* lack apparent sexual dimorphism [24]. The Budgerigar, one of the smallest parrots, displays sexual dimorphism. Male Budgerigar typically feature predominantly blue cere, whereas females tend to have pinkish brown cere. However, these distinctive traits become discernible only as the parakeets reach 6–8 months of age and achieve sexual maturity [22,23].

*Passeriformes* represent the group of birds with the greatest diversity of species—over 6500 in total—and many of them are monomorphic [25]. Passerines are among the most well known of all birds due to their diversity, abundance, and global distribution. Passerines have also played a significant role in human culture and science [26]. While certain passerines exhibit marked sexual dimorphism, others do not exhibit any noticeable differences between sexes. Australian Zebra Finch (*Taeniopygia castanotis*) demonstrates sexual dimorphism. The Gouldian Finch (*Chloebia gouldiae*) showcase sexual dimorphism, with males exhibiting a more vividly colored purple breast band and yellow belly than females. Red Siskin displays pronounced sexual dimorphism, with males exhibiting vibrant red and black colors, while females feature predominantly brown, gray, and black shades with some hints of red. Goldfinch exhibits sexual dimorphism, as evidenced by the red mask extending beyond the eye in males, while in females, it is limited to half the eye. On the other hand, Domestic Canary (*Serinus canaria forma domestica*) does not demonstrate sexual dimorphism. Passerines are most closely linked to parrots (*Psittaciformes*), which are most closely connected to falcons (*Falconiformes*), according to DNA-sequencing research [26].

The order *Accipitriformes* includes 225 species of birds of prey, which are mainly diurnal. They were initially classified in the order *Falconiformes*, but after new genetic research, they were reclassified. Both in *Falconiformes* and in *Accipitriformes*, males are known to be smaller than females, females have larger heads [27], and, in rare cases, plumage is a sexually differentiating feature. Sexual dimorphism does not occur in Eurasian Hobby (*Falco subbuteo*) at an early age, so it would be easier to identify the sex using a molecular method [19].

In the present paper, we used PCR primers (P2/NP) located inside the *CHD1* gene to determine their efficiency in the sex identification of wild and companion birds belonging to six different orders: *Falconiformes*, *Accipitriformes*, *Galliformes*, *Anseriformes*, *Passeriformes*, and *Psittaciformes*. Genetic sexing of birds has many applications in various fields, such as behavioral medicine, conservation medicine, management of wild birds, breeding of different bird species, improvement of breeding programs of captive birds, analysis of breeding strategies of poultry, evolutionary studies, and forensic medicine [3]. Birds are highly social and need pairing in order to increase their welfare. Molecular sexing helps provide welfare elements for birds by early pairing. Mating parrots has been shown to increase their welfare, and, therefore, early sex determination can be of great value to bird owners [28].

The main purpose of this study was to identify the sexes of the above-mentioned wild and companion monomorphic birds (orders *Falconiformes*, *Accipitriformes*, *Galliformes*, *Anseriformes*, *Passeriformes*, and *Psittaciformes*) using PCR techniques and moderately invasively collected samples (oral swabs and feathers) and invasively collected samples (whole blood), provided the owners agreed to the procedure. Despite the observed sexual dimorphism in the tested *Passeriformes* species, our focus was on examining the feasibility of collecting oral swabs from avian species weighing less than 20 g and utilizing exceedingly small feather samples.

## 2. Materials and Methods

### 2.1. Sample Collection

From January to June 2023, samples of feathers, oral swabs, and blood were randomly collected from 43 wild and companion birds as follows: wild birds (*Falconiformes*—Eurasian Hobby, *Accipitriformes*—Common Buzzard (*Buteo buteo*), *Galliformes* (landfowl)—Common Pheasant, and *Anseriformes* (waterfowl)—Whooper Swan) and companion birds (*Passeriformes* and *Psittaciformes*—large-, medium-, and small-sizes parrots) (Table 1). A total of 100 samples (29 paired samples of feathers and oral swabs and 14 tripled samples of feathers, oral swabs, and blood) were taken in this study. All samples were collected during routine check-ups of live birds (*Galliformes, Anseriformes*, *Passeriformes*, and *Psittaciformes*) or cadavers (*Falconiformes* and *Accipitriformes*) admitted to the New Companion Animals veterinary clinic of Faculty of Veterinary Medicine, University of Agricultural Sciences and Veterinary Medicine Cluj-Napoca, Romania.

Two contour feathers with intact calamuses were sampled from the wings or abdominal region of each bird. Oral swabs were collected using sterile cotton swabs (Prima, Taizhou Honod Medical Co., Ltd., Linhai, China) [29]. Blood samples were collected from 10 live birds from the orders *Anseriformes* and *Psittaciformes* through phlebocentesis of the metatarsal veins using heparinized syringes. A total quantity of 0.2–0.3 mL of blood was collected and stored in heparin tubes. Blood samples collected from deceased birds (*n* = 4) from the order *Falconiformes* and *Accipitriformes* were sampled directly from the heart, being identified postmortem in the form of blood clots. The breeders gave consent for all procedures. All samples were collected using surgical gloves, labeled individually, and stored at −20 °C until processing.

### 2.2. DNA Extraction and PCR

Genomic DNA was extracted from all samples (*n* = 100; feathers, oral swabs, and blood) collected from 43 birds. A DNeasy Blood & Tissue Kit (Qiagen, Hilden, Germany) was used according to the manufacturer’s protocol. For DNA extraction from feathers and oral swabs, we used the protocol recommended for tissue samples, while for blood, we used the protocol for nucleated blood. Prior to DNA extraction, the feather calamus was sectioned into small pieces (2–4 mm) and then subjected to mechanical destruction via high-speed shaking with steel beads using a TissueLyserII (Qiagen, Germantown, MD, USA). Oral swabs were transferred to 1.5 mL Eppendorf tubes using sterile scissors. For DNA extraction, we used 25 mg of feather calamus, the entire oral swab, or 10 μL of anticoagulant-treated blood, respectively. DNA concentration was measured using an ND-1000 spectrophotometer (NanoDrop Technologies, Wilmington, DE, USA).

DNA was tested for the presence of the specific genes *CHD1-W* and *CHD1-Z* via conventional PCR, according to the protocol described by Ito et al. [30]. The *CHD1* gene was amplified using the primers P2 (5′-TCT GCA TCG CTA AAT CCT TT-3′) and NP (5′-GAG AAA CTG TGC AAA ACA G-3′) (Generi-Biotech, Hradec Králove, Czech Republic). PCR amplification was performed in a 25 μL reaction mixture consisting of 12.5 μL of MyTaq Red HS Mix (Meridian Bioscience, Cincinnati, OH, USA) and 25 pM of each primer. The total quantity of DNA template added into the reaction mixture was 100 ng. Amplification was performed in a Bio-Rad C1000TM Thermal Cycler (Bio-Rad Laboratories, Hercules, CA, USA). The following cycling conditions were used: initial denaturation at 95 °C for 5 min, followed by 35 cycles consisting of 52 °C for 45 s, 72 °C for 45 s, and 95 °C for 30 s. The amplification was finished with 1 min at 50 °C and a final elongation for 5 min at 72 °C. For sex identification in *Falconiformes* and *Accipitriformes*, a set of three primers (P2, NP, and MP (5′-AGT CAC TAT CAG ATC CGG AA-3′)) was used [30]. PCR amplification was performed under the same conditions as the previous amplification.

Aliquots of each PCR product were electrophoresed on 3% agarose gels stained with RedSafe Nucleic Acid Staining Solution 20,000× (iNtRON Biotechnology, Seongnam, Republic of Korea) and examined under UV light (Bio-Rad BioDoc-ItTM Imaging System, Hercules, CA, USA). The DNA fragment size was compared with a 100 bp DNA ladder (Fermentas; Thermo Fisher Scientific, Waltham, MA, USA), and sex was assigned by counting the visible bands in each lane. Females are characterized by obtaining two bands corresponding to the *CHD1-W* and *CHD1-Z* genes, while males present only one band corresponding to the *CHD1-Z* gene.

## 3. Results

All samples collected from the birds (oral swabs, feathers, and blood) provided a good DNA template for molecular sex identification. The highest DNA concentration was obtained from the blood samples (270.2 ng/μL), followed by the feather samples (74.35 ng/μL) while the lowest concentration was from the oral swab samples (49.82 ng/μL). Identical results were obtained from all types of samples (feathers, oral swabs, and blood) collected from the same birds. Molecular amplification of the *CHD1* gene allowed the identification of the 24 males and 19 females (Table 2). The results of the gel electrophoresis analysis of oral swab samples are presented in Figure 1 and Figure 2.

Since, in the samples collected from Common Buzzard (*Accipitriformes*) and Eurasian Hobby (*Falconiformes*), we failed to identify the sex with the help of the P2 and NP primers (in both males and females only one band appeared), we retested the samples using three primers (P2, NP, and MP), according to the recommendations of Ito et al. [30]. The results obtained for *Accipitriformes* and *Falconiformes* are shown in Figure 3.

## 4. Discussion

In an attempt to find a universal method, several molecular genetic techniques for identifying the sex of birds—as well as many PCR markers based on *CHD1* [2,31,32], *ATP synthase α-subunit* (*ATP5A1*) [33], *W-linked gene for the altered form of protein kinase C-interacting protein* (*Wpkci*) [34], *Nipped-B homolog* (*NIPBL*) [35], *Spindlin* (*SPIN*) [36], or *RAS p21 protein activator 1* (*RASA1*) genes—have been tested [37]. These genes are used to identify differences between the homologous regions of the two Z and W chromosomes based on variations in the length polymorphism of introns located in these regions [38].

In the present study, we tested the efficiency of *CHD1* gene amplification using the P2/NP primer pairs [30] for sex identification in birds from different orders, such as *Accipitriformes*, *Falconiformes, Galliformes*, *Anseriformes*, *Passeriformes*, and *Psittaciformes*. With the exception of *Accipitriformes* and *Falconiformes*, the obtained results showed 100% sex identification in wild and companion birds when using the molecular method based on the intronic length polymorphism. Similar DNA templates were provided for the molecular sexing reactions by all types of samples, including feathers, oral swabs, and blood. Therefore, testing of minimally invasive samples like feathers or oral swabs is adequate to yield a reliable sex identification result in birds. The samples collected minimally invasively such as feathers and oral swabs, although they do not provide a quantity of DNA like the blood samples, have the advantage of minimally handling stress for the birds.

In a previous study, using the pair of primers P2/P8, the PCR success rate of sex identification in birds classified in *Columbiformes* and *Psittaciformes* orders was 94.06% from oral swabs and 82.43% from feathers [13]. According to the results obtained in the present study, and also by Ito et al. [30], the P2/NP primer pair may be used to identify sex in more species than the P2/P8 primer pair, since the P8 primer site is less conserved than the NP primer site.

All the Common Buzzard (*n* = 3) and Eurasian Hobby (*n* = 1) birds included in the present study were cadavers; therefore, sex could be determined via gonad identification. We were unable to determine sex using the P2/NP primers because only a single band appeared in both males and females, probably due to the small difference in size between *CHD1-Z* and *CHD1-W* [30]. According to Nesje and Røed [19], there is just one base difference between the two bands of female Eurasian Hobby. Thus, we retested the samples using multiplex PCR with an additional primer (P2/NP/MP), in accordance with the recommendations of Ito et al. [30]. MP is a 3′-terminal mismatch primer, which allowed the detection of a fragment situated only on the W chromosome [39]. Female-specific *CHD1-W* was detected via NP/MP primers, whereas *CHD1-Z* was amplified by NP/P2 [30]. These primers allowed the successful identification of the sexes of *Accipitriformes* and *Falconiformes* included in the present study. In both species, the females have two bands compared to the males’ single band.

Sex can also be distinguished in birds using a molecular method based on the amplification of a unique sequence located in the W chromosome, regardless of intronic size variation [40]. Therefore, a multiplex PCR, which uses—in addition to the P2/P8 primer pair [2]—a new P0 primer specific for *CHD1-W*, was developed [40]. With the help of this method, several species of birds from 12 avian orders could be sexed, such as *Accipitriformes*, *Galliformes*, *Anseriformes*, *Passeriformes*, and *Psittaciformes* [40].

A new approach to sex determination in birds may be quantitative real-time PCR (qPCR) based on copy number variation of genes associated only with the Z chromosome (*CHRNA6*, *DDX4*, *LPAR1*, *TMEM161B*, and *VPS13A* for neognath species and *DOCK8*, *FUT10*, *PIGG*, and *PSD3* for paleognath species) and absent from the W chromosome [41]. This method has been applied to 73 species with great success, and it has been shown to be a reliable molecular sex identification tool for birds [41].

Until now, even if remarkable research has been performed in the field, no truly universally valid marker or method has been found for the sexing of all bird species. The use of multiple markers is advised for the efficacy of molecular sexing in birds [42] as well as the simultaneous testing of at least two types of minimally invasive samples (feathers and buccal swab).

## 5. Conclusions

The present study demonstrates the applicability of all sample types (feathers, oral swabs, and blood) to determine sex using molecular methods of all examined bird species. Instead of using blood samples, we advise using minimally invasive samples, such as feathers and oral swabs, and testing both types of samples for each bird for accuracy. Molecular sex identification in Red Siskin and Goldfinch has never been reported before, as far as we are aware. Because only a small number of individuals within each species was examined, our study was constrained. Due to the lack of sexual dimorphism in many species and in all nestlings as well as the absence of a universal marker that can be used across all bird species, accurate and effective sexing of birds continues to be challenging.

## Figures and Tables

**Figure 1 animals-13-03417-f001:**
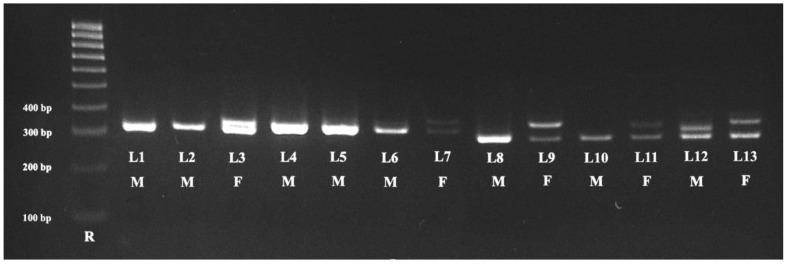
Patterns of the PCR products obtained from *Galliformes* (L1), *Anseriformes* (L2, L3), and *Passeriformes* (L4–L13) using P2/NP primers from oral swab samples. Legend: R—100 bp DNA ladder; L1—Common Pheasant; L2, L3—Whooper Swan; L4, L5—Australian Zebra Finch; L6, L7—Gouldian Finch; L8, L9—Red Siskin; L10, L11—Goldfinch; L12, L13—Domestic Canary. The females (F) presented two bands (L3, L7, L9, L11, and L13), while the males (M) presented a single band (L1, l2, L4–L6, L8, L10, and L12).

**Figure 2 animals-13-03417-f002:**
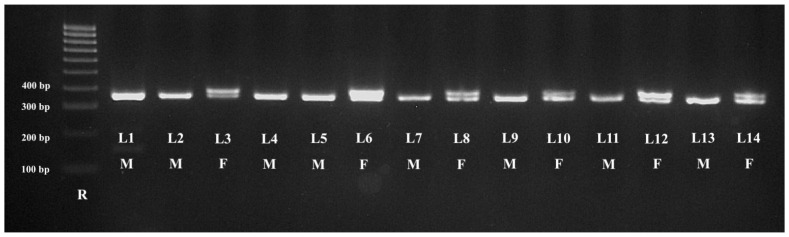
Patterns of PCR products obtained from *Psittaciformes* using P2/NP primers from oral swab samples. Legend: R—100 bp DNA ladder; *Psittaciformes* large size: L1—Scarlet Macaw; L2, L3—African Grey Parrot; L4—White Cockatoo; *Psittaciformes* medium size: L5, L6—Rose-Ringed Parakeet; L7, L8—Red-Rumped Parrot; L9, L10—Cockatiel; L11, L12—Lovebird; *Psittaciformes* small size: L12, L13—Budgerigar. The females (F) presented two bands (L3, L6, L8, L10, L12, and L14), while the males (M) presented a single band (L1, l2, L4, L5, L7, L9, L11, and L13).

**Figure 3 animals-13-03417-f003:**
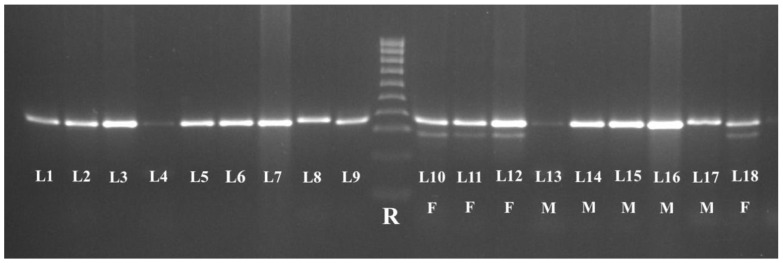
Patterns of PCR products obtained from raptors *Accipitriformes* (L1–L8; L10–L17) and *Falconiformes* (L9; L18) from feather (L1, L4, L10, and L13), oral swab (L2, L5, L8, L9, L11, L14, L17, and L18), and blood samples (L3, L6, L7, L12, L15, and L16). Legend: R—100 bp DNA ladder; L1–L9—P2/NP primer pair was used. A single band was obtained for both sexes. L10–L18—P2/NP/MP primers were used. The females (F) presented two bands (L10–L12 and L18), while the males (M) presented a single band (L13–L17).

**Table 1 animals-13-03417-t001:** Number of samples from oral swabs, feathers, and blood from study birds.

Order	Species	No.	Oral Swabs	Feathers	Blood
*Falconiformes*	Eurasian Hobby (*Falco subbuteo*)	1	1	1	1
*Accipitriformes*	Common Buzzard (*Buteo buteo*)	3	3	3	3
*Galliformes*	Common Pheasant (*Phasianus colchicus*)	2	2	2	-
*Anseriformes*	Whooper Swan (*Cygnus cygnus*)	5	5	5	3
*Passeriformes*	Australian Zebra Finch *(Taeniopygia castanotis*)	2	2	2	-
	Gouldian Finch (*Chloebia gouldiae*)	2	2	2	-
	Red Siskin (*Carduelis cucullata*)	2	2	2	-
	Goldfinch (*Carduelis carduelis major*)	2	2	2	-
	Domestic Canary (*Serinus canaria forma domestica*)	2	2	2	-
*Psittaciformes*	Scarlet Macaw (*Ara macao*)	4	4	4	3
	African Grey Parrot (*Psittacus erithacus*)	4	4	4	2
	White Cockatoo (*Cacatua alba*)	1	1	1	1
	Rose-Ringed Parakeet (*Psittacula krameri*)	2	2	2	1
	Red-Rumped Parrot (*Psephotus haematonotus*)	2	2	2	-
	Cockatiel (*Nymphicus hollandicus*)	3	3	3	-
	Lovebird (*Agapornis fischeri*)	4	4	4	-
	Budgerigar (*Melopsittacus undulatus*)	2	2	2	-
Total		43	43	43	14

**Table 2 animals-13-03417-t002:** Number of males and females determined via molecular methods.

Order	Species	No.	Males	Females
*Falconiformes*	Eurasian Hobby (*Falco subbuteo*)	1	-	1
*Accipitriformes*	Common Buzzard (*Buteo buteo*)	3	1	2
*Galliformes*	Common Pheasant (*Phasianus colchicus*)	2	2	-
*Anseriformes*	Whooper Swan (*Cygnus cygnus*)	5	3	2
*Passeriformes*	Australian Zebra Finch *(Taeniopygia castanotis*)	2	1	1
	Gouldian Finch (*Chloebia gouldiae*)	2	1	1
	Red Siskin (*Carduelis cucullata*)	2	1	1
	Goldfinch (*Carduelis carduelis major*)	2	1	1
	Domestic Canary (*Serinus canaria forma domestica*)	2	1	1
*Psittaciformes*	Scarlet Macaw (*Ara macao*)	4	4	-
	African Grey Parrot (*Psittacus erithacus*)	4	1	3
	White Cockatoo (*Cacatua alba*)	1	1	-
	Rose-Ringed Parakeet (*Psittacula krameri*)	2	1	1
	Red-Rumped Parrot (*Psephotus haematonotus*)	2	1	1
	Cockatiel (*Nymphicus hollandicus*)	3	2	1
	Lovebird (*Agapornis fischeri*)	4	2	2
	Budgerigar (*Melopsittacus undulatus*)	2	1	1
Total		43	24	19

## Data Availability

All the results of the study are presented within the manuscript.

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
