# Peer review of "Minimally Invasive Sampling Methods for Molecular Sexing of Wild and Companion Birds"

_animals, 2023, doi:10.3390/ani13213417_

Round 1
Reviewer 1 Report
Comments and Suggestions for Authors
This paper aims to answer a question that is likely of interest to many other researchers as well as veterinarians and pet-owners. The title, abstract, and introduction all posit to examine the use of 3 different sample materials (blood, oral swabs, and collected feathers) to determine the sex of individual birds. The authors contend that based on their results, all 3 are viable options and that they support the use of the 2 least invasive sample types (oral swabs and feathers) for determining sex compared to traditional methods (laproscopy) or blood sampling. However, there is no data to support the use of all three sample types. The only data shown is derived from an unknown sample type. The authors go on the spend the discussion section to support the use of a new set of primers and how their use allows for sex determination in more avian families compared to previously used primers. While this may all be valid, the issue of different sample types should also be addressed. There is also no data in the results section to support the use of the 3 different sample types and the gels that are pictures give no indication as to which sample types the DNA was extracted from.
Comments on the Quality of English LanguageLine 12: “in nestling the percentage is higher” – in nestlings that percentage is even higher
Line 17: “Conventional PCR was applied in order…” - Conventional PCR was used to amplify the CHD…
Line 22: “molecular sexing helps to provide” – molecular sexing helps provide
Line 27: “from 43 birds were taken into study, as follows…” – included in this study
Line 32: “According to our knowledge…” – To the best of our knowledge…
Line 46: “The sexual determinism…” – Sexual determinism
Line 56-61: This passage is awkward to a native English speaker and I am not sure how to fix it all
Line 63: “thus avoiding the stress of handling wild birds” – thus avoiding the stress of handling.
Line 72: “in order to determine…” – to determine
Line 73: “six different orders, such as…” – six different orders…
Line 114: “taken into de study.” – taken in this study.
Line 137: “For DNA extraction were used 25mg…” – For DNA extraction we used 25mg…
Figures: include sample type in the legends
Line 198: “are used on the purpose to identify…” – are used to identify
Author Response
Dear Reviewer 1,
We appreciate all of the constructive criticisms, useful comments and thoughtful suggestions. All substantive points and suggestions arising from the reviewer have been carefully considered during revision of the original manuscript and implemented (please see below).
All revisions made to the original paper have been highlighted in yellow color.
A detailed response to all comments is outlined in the accompanying letter indicated by sentences starting with AU:. The original paper has been extensively revised in accordance with the comments and recommendations arising from the peer-review process, and as a result is much improved.
We hope that the revisions would allow the manuscript to be considered acceptable for publication.
With our best regards,
Dr. Anamaria Ioana Paștiu
Line 12: “in nestling the percentage is higher” – in nestlings that percentage is even higher AU: Done
Line 17: “Conventional PCR was applied in order…” - Conventional PCR was used to amplify the CHD…AU: Done
Line 22: “molecular sexing helps to provide” – molecular sexing helps provide AU: Done
Line 27: “from 43 birds were taken into study, as follows…” – included in this study AU: Done
Line 32: “According to our knowledge…” – To the best of our knowledge…AU: Done
Line 46: “The sexual determinism…” – Sexual determinism AU: Done
Line 56-61: This passage is awkward to a native English speaker and I am not sure how to fix it all AU: We have changed from „Molecular sexing in birds, based on distinctive characteristics of birds sex chromosomes Z and W, is a non-invasive method, compared to the classical sexing methods, that has many advantages, most important being accuracy and precision. DNA sexing also has economic advantage, given the reduced costs of sample analysis. Molecular sexing is considered a safe method, as collecting samples does not endanger the birds' lives or expose them to risk of infection” in „ Molecular sexing in birds aims to identify distinctive characteristics of sex chromosomes Z and W. Genetic sexing is considered a non-invasive method compared to the classical sexing methods, like for example celioscopy. Moreover, the molecular method is con-sidered safe, as collecting samples does not endanger the birds' lives or expose them to risk of infection.”
Line 63: “thus avoiding the stress of handling wild birds” – thus avoiding the stress of handling. AU: Done
Line 72: “in order to determine…” – to determine AU: Done
Line 73: “six different orders, such as…” – six different orders…AU: Done
Line 114: “taken into de study.” – taken in this study. AU: Done
Line 137: “For DNA extraction were used 25mg…” – For DNA extraction we used 25mg…AU: Done
Figures: include sample type in the legends AU: Done
Line 198: “are used on the purpose to identify…” – are used to identify AU: Done.
Thank you!

Reviewer 2 Report
Comments and Suggestions for Authors
Concerns the manuscript entitled ‘Molecular sexing of wild and companion birds using samples collected by minimally invasive methods’. The manuscript is well written and the language is understandable. However, while the work is interesting, it is not particularly innovative.
Line 119: Change the title of table 1.
Table 1. Number of samples from oral swabs, feathers and blood from study birds.
Line 119: Remove the legend (Legend: * - No. of tested individuals.)
Add one column (as the last one) and label it "Total", instead of the legend (under the table).
Line 124: Why were heparin tubes used?
As we know, heparin is a DNA polymerase inhibitor. EDTA is used as an anticoagulant for genetic analyses. Heparin is used in cytogenetic (not genetic) testing.
What was the concentration of DNA depending on the type of sample (feathers, oral swabs and blood)? This information is missing.
Line 132-133: The DNeasy Blood & Tissue Kit (Qiagen, Hilden, Germany) was used according to the manufacturer's protocol. Protocol modifications are recommended for DNA isolation from avian blood. Red blood cells (erythrocytes) of human (mammalian) blood do not contain cell nuclei and therefore no DNA.
Avian erythrocytes have a cell nucleus. The standard protocol for DNA isolation from blood/tissues is developed for humans (mammals). In the case of DNA isolation from avian blood, modifications to the protocol are necessary. Has the DNA isolation protocol from blood been modified?
Line 145-146: Why did the concentration of DNA template used (from 50 to 270 ng) vary greatly (even by more than 5 times)?
Line 167: Table 2. Number of males and females determined by molecular methods.
Line 168: Remove the legend (Legend: * - No. of tested individuals.)
Add one column (as the last one) and label it "Total", instead of the legend (under the table).
Concerns three figures
Line 170-174:
Line 176-181:
Line 188-191:
No description for the R line.
Why is the molecular weight of DNA denoted as R? It is commonly denoted as M. It should rather be M: 100 bp DNA ladder
Line 246-247: (it is in the text ) The present study demonstrates the applicability of all sample types (feathers, oral swab, blood) for molecular sexing of all examined bird species.
Line 246-247: (should be) The present study demonstrates the applicability of all sample types (feathers, oral swab, blood), to determine the sex using molecular methods of all examined bird species.
Gene names should be in italics. Applies to the entire manuscript.
Author Response
Dear Reviewer 2,
We appreciate all of the constructive criticisms, useful comments and thoughtful suggestions. All substantive points and suggestions arising from the reviewer have been carefully considered during revision of the original manuscript and implemented (please see below).
All revisions made to the original paper have been highlighted in blue color.
A detailed response to all comments is outlined in the accompanying letter indicated by sentences starting with AU:. The original paper has been extensively revised in accordance with the comments and recommendations arising from the peer-review process, and as a result is much improved.
We hope that the revisions would allow the manuscript to be considered acceptable for publication.
With our best regards,
Dr. Anamaria Ioana Paștiu
Line 119: Change the title of table 1.
Table 1. Number of samples from oral swabs, feathers and blood from study birds. AU: Done
Line 119: Remove the legend (Legend: * - No. of tested individuals.)
Add one column (as the last one) and label it "Total", instead of the legend (under the table). AU: Done
Line 124: Why were heparin tubes used?
As we know, heparin is a DNA polymerase inhibitor. EDTA is used as an anticoagulant for genetic analyses. Heparin is used in cytogenetic (not genetic) testing. AU: The reviewer has right regarding that the heparin is a DNA polymerase inhibitor. At the time of collecting the birds blood samples, we did not have EDTA tubes available. We took into account that other authors also used tubes with lithium-heparin for genetic tests from bird blood (Morinha, F., Travassos, P., Seixas, F., Santos, N., Sargo, R., Sousa, L., Bastos, E., 2013. High‐resolution melting analysis for bird sexing: a successful approach to molecular sex identification using different biological samples. Molecular ecology resources, 13: 473-483)
What was the concentration of DNA depending on the type of sample (feathers, oral swabs and blood)? This information is missing. AU: We have added in text: “The highest DNA concentration was obtained in the blood samples (270.2 ng/ml), followed by the feather samples (74.35 ng/ml) while the lowest was in the oral swab samples (49.82 ng/ml).”
Line 132-133: The DNeasy Blood & Tissue Kit (Qiagen, Hilden, Germany) was used according to the manufacturer's protocol. Protocol modifications are recommended for DNA isolation from avian blood. Red blood cells (erythrocytes) of human (mammalian) blood do not contain cell nuclei and therefore no DNA.
Avian erythrocytes have a cell nucleus. The standard protocol for DNA isolation from blood/tissues is developed for humans (mammals). In the case of DNA isolation from avian blood, modifications to the protocol are necessary. Has the DNA isolation protocol from blood been modified? AU: In the DNeasy Blood & Tissue kit (Qiagen) there is described the protocols for tissue, non-nucleated blood as well as for nucleated blood. We used the recommended protocol for nucleated blood, without modifications. We have added in text: “For DNA extraction from feathers and oral swabs we used the protocol recommended for tissue, while for blood we used the protocol for nucleated blood.”
Line 145-146: Why did the concentration of DNA template used (from 50 to 270 ng) vary greatly (even by more than 5 times)? AU: We made a mistake. The DNA concentration, measured by Nanodrop, was between 50-270ng/ml. The highest DNA concentration was obtained in the blood samples (270.2 ng/ml), followed by the feather samples (74.35 ng/ml) while the lowest was in the oral swab samples (49.82 ng/ml). 100 ng of DNA was used as template. We corrected in the manuscript.
Line 167: Table 2. Number of males and females determined by molecular methods. AU: Done
Line 168: Remove the legend (Legend: * - No. of tested individuals.)
Add one column (as the last one) and label it "Total", instead of the legend (under the table). AU: Done
Concerns three figures
Line 170-174:
Line 176-181:
Line 188-191:
No description for the R line.
Why is the molecular weight of DNA denoted as R? It is commonly denoted as M. It should rather be M: 100 bp DNA ladder
AU: We added in legend: R – 100 bp DNA ladder. We keep the notation R (from ruler) to avoid the confusion with M (male).
Line 246-247: (it is in the text ) The present study demonstrates the applicability of all sample types (feathers, oral swab, blood) for molecular sexing of all examined bird species.
Line 246-247: (should be) The present study demonstrates the applicability of all sample types (feathers, oral swab, blood), to determine the sex using molecular methods of all examined bird species. AU: We changed.
Gene names should be in italics. Applies to the entire manuscript. AU: Done
Thank you!
